# Viral Sequences Detection by High-Throughput Sequencing in Cerebrospinal Fluid of Individuals with and without Central Nervous System Disease

**DOI:** 10.3390/genes10080625

**Published:** 2019-08-19

**Authors:** Manuel Schibler, Francisco Brito, Marie-Céline Zanella, Evgeny M. Zdobnov, Florian Laubscher, Arnaud G L’Huillier, Juan Ambrosioni, Noémie Wagner, Klara M Posfay-Barbe, Mylène Docquier, Eduardo Schiffer, Georges L. Savoldelli, Roxane Fournier, Lauriane Lenggenhager, Samuel Cordey, Laurent Kaiser

**Affiliations:** 1Laboratory of Virology, Laboratory Medicine Division, Diagnostic Department, Geneva University Hospitals, 1205 Geneva, Switzerland; 2Swiss Institute of Bioinformatics, 1206 Geneva, Switzerland; 3Department of Genetic Medicine and Development, Faculty of Medicine of Geneva, 1206 Geneva, Switzerland; 4Paediatric Infectious Diseases Unit, Department of Women-Children-Teenagers, Geneva University Hospitals and Medical School, 1205 Geneva, Switzerland; 5Infectious Diseases Service, Hospital Clinic-IDIBAPS, University of Barcelona, 08036 Barcelona, Spain; 6iGE3 Genomics Platform, University of Geneva, 1206 Geneva, Switzerland; 7Anaesthesiology Division, Department of Anaesthesiology, Pharmacology, Intensive Care and Emergency Medicine, Geneva University Hospitals, 1205 Geneva, Switzerland; 8Faculty of Medicine of Geneva, University of Geneva, 1205 Geneva, Switzerland

**Keywords:** acute central nervous system inflammation, meningitis, encephalitis, myelitis, high throughput sequencing, viruses, cerebrospinal fluid, viral sequences

## Abstract

Meningitis, encephalitis, and myelitis are various forms of acute central nervous system (CNS) inflammation, which can coexist and lead to serious sequelae. Known aetiologies include infections and immune-mediated processes. Despite advances in clinical microbiology over the past decades, the cause of acute CNS inflammation remains unknown in approximately 50% of cases. High-throughput sequencing was performed to search for viral sequences in cerebrospinal fluid (CSF) samples collected from 26 patients considered to have acute CNS inflammation of unknown origin, and 10 patients with defined causes of CNS diseases. In order to better grasp the clinical significance of viral sequence data obtained in CSF, 30 patients without CNS disease who had a lumbar puncture performed during elective spinal anaesthesia were also analysed. One case of human astrovirus (HAstV)-MLB2-related meningitis and disseminated infection was identified. No other viral sequences that can easily be linked to CNS inflammation were detected. Viral sequences obtained in all patient groups are discussed. While some of them reflect harmless viral infections, others result from reagent or sample contamination, as well as index hopping. Altogether, this study highlights the potential of high-throughput sequencing in identifying previously unknown viral neuropathogens, as well as the interpretation issues related to its application in clinical microbiology.

## 1. Introduction

Acute central nervous system (CNS) inflammation, encompassing meningitis, encephalitis, myelitis, or any combination of these entities, results either from infection or dysimmunity [1,2]. Among infectious causes, viruses are the main culprits clinicians must search for, once pyogenic bacterial meningitis and nonpyogenic agents, such as *Mycobacterium tuberculosis* or *Listeria monocytogenes* (*Lm*), have been excluded [2,3,4]. Among neurotropic viruses, herpes simplex virus types 1 and 2 (HSV-1 and -2), varicella zoster virus (VZV), enteroviruses (EV), parechovirus, human immunodeficiency virus (HIV), as well as flaviviruses (whose types and prevalence vary according to the local epidemiology) are the predominant causes [5,6]. 

Despite significant improvements in microbiological investigations in the past decades through implementation of sensitive PCR-based assays performed on cerebrospinal fluid (CSF) or on cerebral biopsy samples, the cause of acute meningo-encephalitis remains unknown in approximately 50% of cases [2,3]. As the initial clinical features of this syndrome are often unspecific and cannot assert an aetiology, a large panel of molecular assays that are by essence formatted to screen predefined targets is often used. High-throughput sequencing (HTS) represents a tool enabling an unbiased microbial detection [7,8].

In the present investigation, HTS was used to screen for RNA and DNA viral signatures in CSF withdrawn from both patients admitted and investigated for acute CNS inflammation and in a control group of individuals undergoing spinal anaesthesia for an elective surgical procedure. This strategy intended not only to detect unexpected or divergent viral neuropathogens, but also to explore the human CNS virome.

## 2. Materials and Methods

The protocol of this study was approved by the Geneva Cantonal Ethics Commission (project numbers #13-074 and #2016-00549). Patients or their next of kin provided written informed consent before enrolment.

This study is a monocentric prospective unmatched case-control study performed at the Geneva University Hospitals, Switzerland. Inclusion criteria were: paediatric and adult patients hospitalized at Geneva University Hospitals from May 2013 through June 2017, presenting with suspected CNS inflammation, and for which no diagnosis was made in the first 48 h. Exclusion criteria were: the absence of a signed informed consent for the present study. 

CNS inflammations were classified as three entities: encephalitis/meningoencephalitis, meningitis, and myelitis/meningomyelitis (see Table 1 for definitions) [9,10,11].

Concerning the control group, inclusion criteria were: adult patients without any known CNS disease, hospitalized from January 2016 through December 2017 either for elective surgery (orthopaedic, urologic, or visceral) under spinal anaesthesia; or women who underwent elective caesarean section under spinal anaesthesia. Exclusion criteria were: the absence of a signed informed consent for the present study. 

CSF samples were collected through lumbar puncture performed for diagnostic purposes in the case group and during spinal anaesthesia, before injecting the anaesthetic mixture, in the control group. In the event that several CSF samples were collected from the same patient, only the first was selected for analysis. No patient received antiviral treatment before lumbar puncture except one immunocompromised patient treated with valaciclovir for secondary prevention of a herpes zoster infection, and an HIV-infected patient treated with atazanavir and ritonavir. All samples were stored at −80 °C before HTS analysis.

For all patients with CNS disease, a workup according to the standard of care was performed for diagnostic purposes. All CSFs were screened for EV, HSV-1 and -2, and VZV by in-house real-time (reverse transcription)-PCR rRT-PCR assays until June 2017, when the FilmArray Meningitis/Encephalitis (ME) Panel [12] was introduced as a routine tool to screen CSF for pathogens. Serological screening for HIV (*n* = 32/36), syphilis (*n* = 29/36), and Lyme disease (*n* = 33/36) was also performed in most patients. A cerebral CT scan was performed for all patients having encephalitis or myelitis. Additional microbiologic investigations, as well as radiological evaluation by CT-scan and/or MRI, electroencephalogram, immunological and oncologic investigations, were performed on a case-by-case basis.

In the control group, no analysis other than HTS on CSF samples was performed.

Medical records of the patients with CNS disease (*n* = 36) and controls (*n* = 30) were collected using a standardized CRF that included the main patients’ characteristics, and for patients with CNS disease, clinical, laboratory, and radiological data.

The viral enrichment process, the viral nucleic acid extraction, and the library preparations were done using two specific protocols for RNA and DNA genome viruses as previously described [13]. Briefly, two hundred microliters (μL) of each sample were treated with 40 U of Turbo DNAse (Ambion, Rotkreuz, Switzerland). Then, half the volume (120 μL) was used to perform RNA extraction using the TRIzol protocol (Invitrogen, Carlsbad, CA, USA), while the second half was used to perform a DNA extraction with a NucliSens easyMAG system (bioMérieux, Geneva, Switzerland), followed by a double-stranded DNA synthesis step (Klenow, New England BioLabs, Ipswich, MA, USA). Concerning the RNA sequencing library preparation, ribosomal RNA was removed prior to the preparation of the RNA libraries with the TruSeq total RNA preparation protocol (Illumina, San Diego, CA, US). DNA libraries were prepared using the Illumina Nextera XT protocol from Illumina. Then, concentrations of both RNA and DNA libraries were measure with a Q-bit (Life Technologies, Carlsbad, CA, USA) and the size distribution of fragments was checked with a 2200 TapeStation (Agilent, Santa Clara, CA, USA). For samples drawn from patients with CNS disease, RNA and DNA libraries were both run on a HiSeq 2500 (Illumina, San Diego, CA, USA; paired-end sequencing, 100-bp protocol). For the 30 control samples, RNA and DNA libraries were run on a HiSeq 2500 and 4000, respectively (Illumina, paired-end sequencing, 100-bp protocol). Raw data were analysed using an updated version of the ezVIR pipeline [13]. A ratio of 0.3% was used for cross-talk check [14]. In parallel, a de novo analysis was done on nonhuman reads using the assembly IDBA-UD software (v.1.1.3) [15]. Contigs > 1000 bp were blasted (blastx BLAST 2.3.0+) against a clustered version of the RVDB-prot 12.2 (http://rvdb-prot.pasteur.fr/) protein database.

In order to confirm HTS findings or confirm suspected sequence cross-lane, a set of specific rRT-PCR assays were applied to CSF with a sufficient initial specimen leftover to extract the viral genomes using the NucliSENS easyMAG (bioMérieux, Geneva, Switzerland). Such assays were performed when needed and available for adenovirus (ADV), Torque teno virus (TTV), Epstein-Barr virus (EBV), Merkel cell polyomavirus (MCPyV), human pegivirus-1 (HPgV-1), and Human herpesvirus 7 (HHV-7) (Appendix A) [16,17,18,19,20]. The rRT-PCR assays were performed using the one-step QuantiTect Probe RT-PCR Kit (Qiagen, Hombrechtikon, Switzerland) in a StepOne Plus instrument (Applied Biosystems, Rotkreuz, Switzerland) for the HPgV-1 assays or using the TaqMan™ Universal PCR Master Mix (ThermoFisher, Reinach, Switzerland) on a QuantStudio 5 instrument (Applied Biosystems) for the others assays.

Data from atients and controls were described as frequency and percentage for categorical parameters and as mean ± standard deviation (SD) for continuous parameters. Comparisons between groups were performed using the Mann–Whitney test for continuous variables. A two-sided *p*-value of <0.05 was considered significant. Statistics were performed using Stata (StataCorp. 2015, College Station, TX, USA).

## 3. Results

### 3.1. Clinical and Paraclinical Data

A total of 36 patients with suspected clinical acute CNS inflammation were included, and 30 patients without any CNS symptoms were included in the control group. Among the 36 enrolled cases with suspected CNS disease, a definitive diagnosis using routine investigations was made in 10 of them (nine adults and one child), who were collectively called the “diagnosed CNS disease” group. One patient had *Lm* meningo-encephalitis, one had an HIV-associated encephalitis, and eight had a documented noninfectious cause of CNS disease (Figure 1). The remaining 26 cases (23 adults and three children) composed a group with “acute CNS inflammation of unknown origin” and presented clinical signs compatible with acute encephalitis (*n* = 2), acute meningitis (*n* = 12), meningo-encephalitis (*n* = 7), myelitis (*n* = 1), and meningo-myelitis (*n* = 4) (Figure 1).

Demographic characteristics for patients and controls are shown in Table 2 and Table 3, respectively.

Three patients with CNS disease (8.3%) were under immunosuppressive treatment (one liver and one kidney transplant recipients, and one patient with systemic lupus erythematosus).

The 26 cases with meningitis, encephalitis, and/or myelitis of undetermined cause were classified into three groups according to the definitions proposed in Table 1. Clinical and laboratory data are described in Table 4. Regarding CSF analysis, the patients’ median white blood cell count at the time of the CSF collection was 76 M/L (range 6–3706 M/L), most of the patients had proteinorachia above 0.45 g/L (range 0.24–2.32 g/L), and median glycorachia was 3.1 mmol/L (range 2.5–5.4 mmol/L) (Table 4). The median CRP level was 4.3 mg/L (range 0–266 mg/L) and median peripheral blood leucocyte count was 10 G/L (range 3.9–25.1 G/L) (Table 4).

Among the 26 patients with CNS inflammation of unknown cause using routine investigations, the time between onset of neurologic symptoms and lumbar puncture was available for 23 patients, and the median value was 3 days (range 0–8 days).

### 3.2. HTS Analysis and r(RT-)PCR Results

Due to the use of different HTS platforms over time for the DNA libraries and multiplexing protocols between cases and controls (see methods), the total million paired reads were lower in the samples from patients without CNS disease (control group) compared to those of patients with CNS inflammation and other CNS diseases (Appendix A). In order to confirm the sensitivity of the HTS method despite this variability, screening TTV with quantitative real-time PCR of all samples from patients without CNS disease was performed and revealed to be positive in one CSF (CSF31, viral load < 250 copies/mL), leading to concordant results with HTS and PCR assay in 29/30 (96.7%) samples.

#### 3.2.1. Detection of Nonsignificant Viral Sequences and Reagent Contamination 

Patients undergoing CSF puncture for anaesthesia purposes were considered as asymptomatic controls (*n* = 30) and a negative control for the whole HTS procedure was used. This led us to identify viral sequences reflecting HTS reagents contaminants (Appendix A).

Among the 12 MCPyV sequences identified in this group, none were confirmed by rPCR (Appendix A).

Papillomavirus and Polyomavirus sequences were obtained in 26/30 and 9/30 samples, respectively, as well as in the negative control. Although no specific rPCR assay was available for these two virus families, these findings most likely indicate environment or reagent contaminations.

Reagent contaminations with vector sequences were detected in three samples. Two were positive for adenovirus (CSF36 and CSF37) and one for parvovirus (CSF12) sequences.

#### 3.2.2. Sequence Cross-Contamination

In 1/9 samples positive for anelloviruses by HTS (P36), reads were falsely assigned because of sequence cross-contamination (Appendix A).

#### 3.2.3. Viral Sequences Assigned to Pathogenic or Commensal Viruses

##### Control Patients

Anelloviruses were not detected in the CSF samples collected from the 30 controls. 

HPgV-1 sequences detected by HTS in two CSF samples were confirmed by rRT-PCR in samples CSF33 and CSF37 (Appendix A).

Two samples tested positive for HHV-7 sequences by HTS. One of these was considered to result from index hopping. None of these two samples was confirmed by rPCR (Appendix A).

Gemycircularvirus (GemyCV) sequences were detected by HTS in one sample, and Molluscum contagiosum virus sequences in another sample in this group. These findings could not be confirmed due to the lack of specific rPCR assays for these viruses in our laboratory (Figure 2, and Appendix A).

##### Patients with CNS Inflammation of Unknown Origin

In this group, HTS analysis allowed the identification of a potential viral causal agent in one case only. Indeed, the presence of novel human astrovirus (HAstV)-MLB2 sequences (155 reads, 35% genome coverage) were detected in the CSF of an immunocompetent patient, as well as in the plasma, anal swab, and urine samples retrospectively tested. This virus, which is not screened for in a routine work-up, was considered to be associated with the patient’s clinical presentation and a final diagnosis of HAstV-MLB2-associated meningitis was retained [21].

Anelloviruses’ reads were detected in eight of the 26 patients with acute CNS inflammation of unknown origin (26.9%). A specific rPCR assay confirmed the detection of TTV DNA in 7/8 samples (87.5%) (Appendix A).

Of note, the mean lymphocyte count of the seven TTV positive CSF samples (524.4 G/L, SD 1068.3 G/L) was higher compared to the 19 TTV negative CSF samples (mean 65.7 G/L, SD 54.7 G/L), although statistical significance was not achieved (*p* = 0.201).

An EBV sequence detected by HTS was not confirmed by an EBV rPCR assays (Figure 2, and Appendix A).

Megavirus sequences were detected by HTS in one sample, and GemyCV sequences in another one. These findings could not be confirmed due to the lack of specific rPCR assays for these viruses in our laboratory (Figure 2, and Appendix A).

##### Patients with Diagnosed CNS Disease

In this group, HPgV-1 sequences were detected by HTS in one patient, and were confirmed by rRT-PCR (Appendix A).

In one of the four samples in which MCPyV sequences were detected by HTS, MCPyV DNA detection was confirmed by rPCR (Appendix A).

Anelloviruses’ reads were detected in one CSF sample. This finding could not be confirmed by rPCR due to insufficient leftover sample volume.

HHV-7 sequences were detected by HTS but not by rPCR in one sample (Figure 2, and Appendix A).

GemyCV sequences were detected by HTS in one sample. This finding could not be confirmed due to the lack of a specific rPCR assay in our laboratory (Figure 2, and Appendix A).

Of note, in a case of HIV escape in the CNS diagnosed by rRT-PCR (110 RNA copies/mL), HIV sequences were not detected by HTS (Appendix A), possibly due to the low viral load and limited amount of CSF analysed.

#### 3.2.4. De Novo Analysis of the HTS Data

Finally, a de novo analysis of the HTS data did not reveal the presence of other relevant viral sequences.

## 4. Discussion

An original approach of the present investigation was the inclusion of a control group of patients without CNS disease, providing the opportunity to explore the CSF virome of asymptomatic humans. This study reports the HTS analysis results of CSF samples collected from 26 patients presenting with CNS inflammation of unknown origin, 10 patients with various identified causes of CNS disease, and 30 control adult patients.

Among the 26 patients with CNS inflammation, an unexpected viral pathogen was identified by HTS and considered to be responsible for the CNS disease in only one case, reported by Cordey et al. [21]. HTS led to the identification of other sequences assigned to viruses, some of them known to be associated with human infection and diseases but their causal role in CNS inflammation in our cases cannot be ascertained. HPgV-1 was detected in the CSF of two patients without CNS disease and one adult patient with CNS HIV escape. HPgV-1 has for the first time been detected in CNS using HTS on a brain sample of a patient with multiple sclerosis [22] and with RT-PCR in CSF samples from HIV infected patients, and its detection could possibly be linked to concomitant viremia [23,24]. Thus far, HPgV-1 detection in CNS samples has not been associated with any overt disease. TTV detection has been reported in CSF samples of patients with infectious and noninfectious CNS diseases, including patients with concomitant viremia [25,26,27]. Since TTV is known to be lymphocytotropic [28,29], the correlation between high CSF lymphocyte counts and increased TTV DNA detection [28,29] suggests that TTV enters the CSF in situations where the blood–brain barrier is altered, either associated to lymphocytes or as free circulating virus [30]. Furthermore, TTV was not detected in any CSF from patients without CNS disease by HTS or by rPCR.

GemyCV detection in CSF samples in both cases (two samples) and controls (one sample) reflects the uncertainty about the origin of GemyCV in CSF samples and its potential role in CNS disease [27,31,32]. Although no GemyCV sequences were detected in the negative control, these viruses, which are known to infect fungi, are not expected to cause human diseases and their detection could be associated with punctual environment or reagent contamination [33].

Papillomavirus, MCPyV, and other polyomavirus sequences were detected with HTS in CSF samples of both patients with and without CNS disease, as well as in the HTS whole process negative control (Figure 2), suggesting HTS contaminations. As mentioned earlier, real-time PCR results for MCPyV were all negative except in one case, a liver transplant recipient. Data are conflicting regarding the detection of MCPyV DNA in different types of brain tumours and its detection in cases with CNS infections has not been reported [34,35,36]. 

Sequence cross-contaminations were considered for two samples positive for HHV-7 (CSF23 and P33). Indeed, the detection of HHV-7 specific reads by HTS but not confirmed by real-time PCR likely results from experimental cross-contaminations during the library preparation with another clinical specimen prepared in the same series but that is not part of the present study. Although we cannot rule out that in these latter cases HTS was more sensitive that the specific real-time PCR, these results highlight that this issue must systematically be considered. Furthermore, in three healthy individuals, reagents contaminations with known vector sequences were also observed for two adenovirus and one parvovirus positive samples. 

The paucity of relevant viral sequences detection by HTS in CSF collected in controls but also from patients who have CNS inflammation of undetermined aetiology may be explained by the following hypotheses. First, some viruses are known to be undetectable in CSF by the time of CNS disease. This is the case of tick-borne encephalitis virus (TBEV), whose associated CNS disease is probably rather related to the host immune and inflammatory response, rather than to tissue damage resulting from uncontrolled viral multiplication. The same phenomenon might apply for other neurotropic viruses. Second, the time between onset of neurologic symptoms and lumbar puncture might be too long in some cases, although this time interval was relatively short in our study (median 3 days, range 0–8 days). In such instances, serology represents an essential complementary diagnostic tool. Third, for some viral CNS infections, the microbiologic diagnostic yield might be higher in samples other than CSF. This is particularly true for enteric viruses, for which nucleic acid detection can be more important in terms of quantity and more prolonged in stool samples or anal swabs than in CSF. Fourth, a large, potentially under-recognised proportion of encephalitis and myelitis might result from autoimmunity, either related to post infectious, paraneoplastic, or idiopathic processes. Fifth, drug-related aseptic meningitis might also account for a portion of negative CSF HTS results. Finally, while being the ideal sample for investigating meningitis, CSF is only a suboptimal surrogate in cases of brain parenchymal infection. 

The enrolment of patients with an initial suspicion of CNS inflammation started in 2013 and the respective CSF samples were consecutively analysed by HTS, generating a large number of runs. Therefore, given costs constraints (although since HTS-related costs have gradually decreased), a shortcoming of the study is the absence of positive (positive sample or spiked internal RNA/DNA controls) and negative controls run in parallel to evaluate the analysis efficiency (such as sensitivity loss, potential detection of reagent, or environmental contaminations). A HTS whole-process negative control was however included for CSF samples collected from patients without CNS disease. Finally, although our pipeline has been challenged on different specimens positive for a wide range of RNA and DNA viruses with an overall success rate suggesting a sensitivity threshold close to that of rRT-PCR assays [13,21,37,38], and benchmarked to the VirCapSeq-VERT pipeline, which is based on a positive selection of the viral template prior to sequencing using a probe set [39], with similar results obtained, we cannot exclude that some viral sequences were not detected by our unbiased method.

Concerning the control group, the results of HTS and confirmatory rRT-PCR assays revealed the absence of relevant confirmed viral sequences, except HPgV-1 sequences in two CSF samples. This information will be useful for future interpretations of HTS results in samples from patients presenting with CNS inflammation.

Regarding the case group, the low diagnostic yield could probably be interpreted according to the multiple factors described above. Nonetheless, the added value of HTS in the etiologic identification of a viral cause of CNS infections has been demonstrated [7,8]. HTS will undoubtedly be implemented in routine clinical virology laboratories as a second line technique or in parallel to an extensive work up driven by guidelines and local epidemiology. Special caution should be taken when interpreting HTS results in the context of CNS disease, due to possible environmental or reagent contaminations, sequence cross-contaminations, detection of viral latency, and identification of nonpathogenic viral bystanders, as illustrated in this study and recently reported data [33].

## Figures and Tables

**Figure 1 genes-10-00625-f001:**
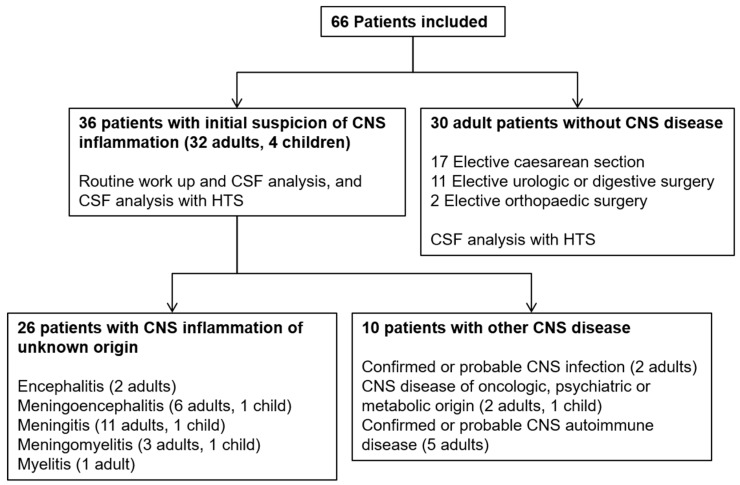
Study flowchart showing patient groups included in the study.

**Figure 2 genes-10-00625-f002:**
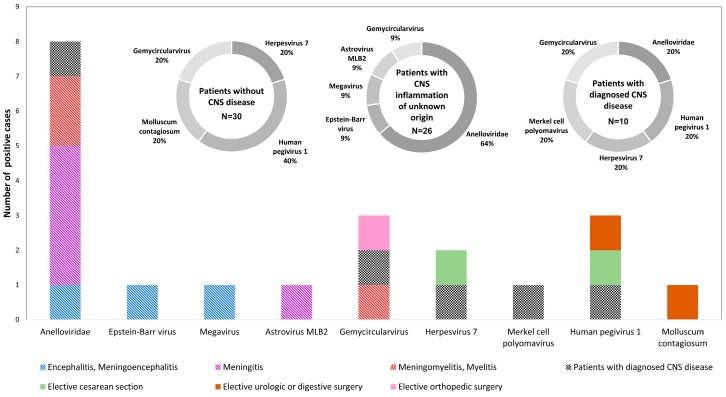
Schematic representation of the different RNA and DNA viruses detected by high throughput sequencing (HTS) in patients with and without CNS disease.

**Table 1 genes-10-00625-t001:** Acute central nervous system (CNS) inflammation definitions.

Encephalitis and Meningo-Encephalitis	Meningitis	Myelitis and Meningomyelitis
Major criteria (required) Altered mental status (defined as decreased or altered level of consciousness, lethargy or personality change) lasting ≥24 hours with no alternative cause identified Minor criteria (2 required for possible encephalitis, ≥3 required for probable, or confirmed encephalitis) Documented fever ≥ 38.2 °C within the 72 h before or after presentation Generalized or partial seizures not fully attributable to a preexisting seizure disorder New onset of focal neurologic findings CSF leukocyte count ≥ 5 M/L Abnormality of brain parenchyma on neuroimaging suggestive of encephalitis Abnormality on EEG that is consistent with encephalitis and not attributable to another cause Note: Diagnosis of meningo-encephalitis is considered if clinical, radiological or biological findings are suggestive of meningeal inflammation	≥2 following criteria Headache Fever ≥ 38.2°C Photo- and/or Phonophobia Neck stiffness **AND** CSF leukocyte count ≥5 M/L	Major criteria (≥1 required) Asymmetrical flaccid weakness with reduced or absent reflexes or sensory symptoms or signs Hyper intensities of the spinal cord on T2 weighted MRI imaging Minor criteria (optional) Fever ≥ 38.2 °C CSF leukocyte count ≥5 M/L Note: Diagnosis of meningo-encephalitis is considered if clinical, radiological or biological findings is suggestive of meningeal inflammation

Encephalitis definition was based on [11]; myelitis definition was adapted from [9,10].

**Table 2 genes-10-00625-t002:** Demographic characteristics of patients with central nervous system (CNS) disease.

	All	CNS Inflammation of Unknown Origin	Diagnosed CNS Disease
**Number (%)**	36 (100)	26 (72.2)	10 (27.8)
**Male**	22 (61.1)	16 (61.5)	6 (60)
**Mean age, years (SD)**	41.6 (18.5)	38.7 (18.5)	49.2 (17.1)
**<18 years old (%)**	4 (11.1)	3 (11.5)	1 (10.0)

**Table 3 genes-10-00625-t003:** Demographic characteristics of the control patients.

	All	Urologic or Digestive Surgery	Orthopaedic Surgery	Caesarean Section
**Number (%)**	30 (100)	11 (36.6)	2 (6.7)	17 (56.7)
**Male**	9 (30)	9 (81.8)	0	0
**Mean age, years (SD)**	53.4 (22.5)	77.4 (7.5)	81 (1.4)	34.7 (4.1)

**Table 4 genes-10-00625-t004:** Clinical and laboratory data of the 26 patients with CNS inflammation of unknown origin.

	All (*N* = 26)	Encephalitis and Meningo-encephalitis (*n* = 9)	Meningitis (*n* = 12)	Myelitis and Meningo-myelitis (*n* = 5)
**Clinical features, *n* (%)**				
**Headache**	19 (73.1)	5 (55.6)	11 (91.7)	3 (60.0)
**Neck stiffness**	3 (11.5)	2 (22.2)	1 (8.3)	0
**Fever (≥38,2 °C) *^1^**	13 (50.0)	3 (33.3)	7 (58.3)	3 (60.0)
**Nausea/vomiting**	2(7.7)	0	2 (16.7)	0
**Photophobia**	8 (30.8)	2 (22.2)	5 (41.6)	1 (20.0)
**Phonophobia**	4 (15.4)	0	3 (25.0)	1 (20.0)
**Altered mental status *^2^**	8 (30.8)	8 (88.9)	0	0
**Seizure**	2 (7.7)	2 (22.2)	0	0
**Suggestive abnormality on neuroimaging**	13 (50.0)	5 (55.5)	4 (33.3)	4 (80.0)
**Suggestive abnormality on EEG *^3^**	8 (30.8)	7 (77.8)	1 (8.3)	0
**Sensory or motor focal neurologic deficit**	9 (34.6)	0	4 (33.3)	5 (100.0)
**Requiring intensive care**	4 (15.4)	2 (22.2)	0	2 (40.0)
**Requiring immunosuppressive therapy**	6 (23.1)	1 (11.1)	1 (8.3)	4 (80.0)
**Clinical outcome *^4^**				
**Complete resolution of signs and symptoms**	16 (61.5)	3 (33.3)	11 (91.6)	2 (40.0)
**Death**	0	0	0	0

**Laboratory features**				
**Cerebrospinal fluid**				
**WBC *^5^ count ≥ 5 M/L (%)**	26 (100.0)	9 (100.0)	12 (100.0)	5 (100)
**Median WBC count, M/L (range)**	76 (6–3706)	53 (18–72)	75(6–3706)	161 (50–275)
**Median glycorachia, mmol/L (range) (2.8–4.0 mmol/L)**	3.1 (2.5–5.4)	3.2 (2.7–5.4)	3.2 (2.7–4.5)	2.8 (2.5–4.4)
**Median proteinorachia (g/L) (range) (0.15–0.45 g/L)**	0.61 (0.24–2.32)	0.7 (0.37–2.32)	0.61 (0.24–2.27)	0.53 (0.40–1.58)
**Proteinorachia > 0.45 (%)**	21 (80.8)	8 (88.9)	9 (75.0)	3 (60.0)
**Proteinorachia < 0.15 (%)**	0	0	0	0
**Blood**				
**Median C-reactive protein mg/L, (range) (0–10 mg/L) *^6^**	4.3 (0.0–266.1)	1.7 (0.0–80.8)	4.8 (1.6–266.1)	18.1 (7.7–33.7)
**Median Leucocytes G/L,** **(range) (4–11 G/L) *^6^**	10.0 (3.9–25.1)	7.4 (3.9–13.9)	10.9 (4.9–25.1)	14.4 (6.9–15.1)

^*1^ Documented fever within 72 h before or after presentation to medical attention, ^*2^ cf. Table 3, ^*3^ electroencephalogram, ^*4^ within the three months after the first day of hospitalization, ^*5^ white blood cells, ^*6^ Laboratory reference ranges.

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
