# Peer review of "Viral Sequences Detection by High-Throughput Sequencing in Cerebrospinal Fluid of Individuals with and without Central Nervous System Disease"

_genes, 2019, doi:10.3390/genes10080625_

Round 1
Reviewer 1 Report
This paper is one of several recent studies evaluating the use of metagenomic NGS for the diagnosis of CNS infection in patients with evidence of meningitis, encephalitis and myelitis.
It is a well-written, considered account by infectious diseases specialists who are expert in their field and while the study is not entirely novel, it will be of high interest to readers interested in the application of MNGS to the diagnosis of CNS infection. I liked that the authors highlighted cross-contamination as a difficulty - it is universal and needs to be controlled and quantified but cannot be eradicated completely.
I have a number of questions
The sequencing was done at huge depth and I am perhaps a little surprised that the authors didn't detect more viruses. What was missing entirely from the results section was the positive control data from the known diagnosis patients. Was this successful? If so, then perhaps as the authors suggest, the samples were perhaps selected from patients a bit too late to detect nucleic acid (I'd have thought 8 days might be too late for several viruses as the authors suggest. The presence of fever of 38.2 degrees in only 50% might also indicate that the sampling was carried out a little too late). It would be reassuring to know that the positive control viruses were detected and what the CT values or viral load was for these. Otherwise, perhaps there is an issue with the sequencing method (certainly not the depth, but perhaps other aspects of the wet lab or bioinformatic method). For example, the method used in the recent Wilson paper cited by the authors is to deplete human DNA with anti-methylation Ab. Alternatively, the authors could have enriched for viruses for example using a method such as VirCap. I think some thought/discussion around this could improve the paper.
The bioinformatic methods are fairly well described in the cited paper although a short summary (a couple of sentences) regarding this method would have been helpful.
I'd have liked to see %coverage of the genome for the astrovirus and the accession number for GenBank
Minor issue - I believe human pegivirus 1 should be abbreviated as HPgV-1.
Author Response
We should like to thank reviewer 1 for the constructive comments and the appreciation of the work.
The sequencing was done at huge depth and I am perhaps a little surprised that the authors didn't detect more viruses. What was missing entirely from the results section was the positive control data from the known diagnosis patients. Was this successful? If so, then perhaps as the authors suggest, the samples were perhaps selected from patients a bit too late to detect nucleic acid (I'd have thought 8 days might be too late for several viruses as the authors suggest. The presence of fever of 38.2 degrees in only 50% might also indicate that the sampling was carried out a little too late). It would be reassuring to know that the positive control viruses were detected and what the CT values or viral load was for these. Otherwise, perhaps there is an issue with the sequencing method (certainly not the depth, but perhaps other aspects of the wet lab or bioinformatic method). For example, the method used in the recent Wilson paper cited by the authors is to deplete human DNA with anti-methylation Ab. Alternatively, the authors could have enriched for viruses for example using a method such as VirCap. I think some thought/discussion around this could improve the paper.
We thank the reviewer for these pertinent comments. We fully understand the reviewer’s frustration not to detect more viruses. However, it unfortunately reflects the HTS reality in CSF samples as shown in our recently published review (HTS led to the identification of a viral cause in 13% of CNS infections cases in which routine assays failed to identify the cause) (Zanella MC et al. Clin Microbiol Infect. 2019 Apr;25(4):422-430).
We agree that whole-process positive controls are recommended (i.e. spiked samples or positive samples run in parallel). Regarding the “positive controls” mentioned by the reviewer, i.e. the patients in which an infectious causes was identified via conventional microbiology diagnostics, we only had two such cases. One had Listeria monocytogenes meningoencephalitis, diagnosed by bacterial culture. Since our HTS pipeline is has been designed for viral genome sequencing (from the nucleic extraction methods to bioinformatics analysis), this latter patient could not be considered as whole-process positive control. The other patient had compartmentalized CNS HIV replication, but with low viral RNA load, as assessed by a commercial real-time RT-PCR assay (100 copies/ml). No HIV sequence was detected by HTS for this latter, which further confirms that HTS remains frequently slightly less sensitive than real-time (RT-)PCR (independently of the sequencing depth used) that can be affected by different factors, such as DNA or RNA background (Wilson MR et al. N Engl J Med. 2019 Jun 13;380(24):2327-2340). However, our HTS pipeline has been validated on a large panel of clinical specimen positive for a wide range of RNA and DNA viruses (Petty TJ et al. J Clin Microbiol. 2014 Sep;52(9):3351-61) with excellent overall sensitivity observed compared to routine real-time (RT-)PCR assays, and in some case with an even better specificity (Cordey S et al. Clin Microbiol Infect. 2015 Apr;21(4):387.e1-4). Furthermore, our unbiased HTS method was recently benchmarked with VirCapSeq-VERT a pipeline based on a positive selection (i.e. probe set) of the viral template prior to sequencing. Although VIRCapSe-VERT required a lower number of reads (our unbiased approach provided a better coverage of genome termini) similar results were obtained by both pipelines (Williams SH et al. mSphere. 2018 Aug 22;3(4). pii: e00311-18). As requested by the reviewer, this point is now discussed in the revised version of the manuscript. Finally, we agree that a limitation to our study is the absence of positive HTS procedure controls. This is now mentioned in the revised version.
The bioinformatic methods are fairly well described in the cited paper although a short summary (a couple of sentences) regarding this method would have been helpful.
These methods are now summarized in a paragraph.
I'd have liked to see % coverage of the genome for the astrovirus and the accession number for GenBank
As requested, the number of specific reads detected in the CSF sample as well as the % genome coverage are now mentioned in the revised version.
The HAstV-MLB2 sequence was not deposited in GenBank but showed 98.5% nucleotide sequence identity homology with the complete genome of the isolate MLB2/human/Stl/WD0559/2008. Please refer to Cordey S. et al Emerg Infect Dis. 2016 May; 22(5): 846–853 for more details.
Minor issue - I believe human pegivirus 1 should be abbreviated as HPgV-1.
This misspelling has been corrected throughout the manuscript.
Reviewer 2 Report
Manuel et al. investigate the use of high-throughput sequencing (HTS) for direct identification of viral etiologies in individuals with or without central nervous system disease. This manuscript presents interesting findings which may add to the growing understanding of clinical utilization of NGS/HTS to aid diagnosis of infections, particularly in CNS infections with unknown causes. Of note, the inclusion of 30 specimens from patients with no known CNS disease provides the opportunity to explore the CSF virome. However, this manuscript may be benefit from revision. I would like to encourage the authors to address the comments and resubmit the manuscript.
Major comment1: Line 83, I would prefer the authors to clarify how positive control (monitor the nucleic acid recovery) and negative control (reagent control) were performed in this study and how these controls were used to help HTS analysis.
Major comment 2: Line 99, the cell/chemistry analysis for the CSFs from the control group would add valuable information and help HTS data analysis from this group.
Major comment 3: Line 134-135, please indicate if HTS identified these two infectious agents and how HTS results correlate with the diagnosis in these two cases.
Major comment 4: Section 3.5,
The viral sequences assigned between two groups should be presented separately and included in the text so that readers would have a better overview.
The data from each virus should be present in one paragraph/section to avoid confusion: For example, both line202 and 204 mentioned MCPyV, same as in line 204 and 206 for HpgV-1.
Line 199 and 200, the numbers of TTV positive and negative sample should be included.
Line 205, should be Figure S2, panel C
Major comment 5: The authors used specific r(RT)-PCR or RT-PCR assays to evaluate the trueness of the HTS findings (in line 113) and considered the identification of HHV7 from one control CSF as contamination (line 256). I would like to see the discussion of the rationale and compare the performance of HST vs. RT-PCR and cite relevant references.
Minor comment 1: Table 1, the bullets are not lined up.
Minor comment 2: Table 4, please include the normal ranges for the glucose and protein.
Minor comment 3: Figure 2, please include the total number of samples for each group.
Author Response
We should like to thank reviewer 2 for the constructive comments and the appreciation of the work.
Manuel et al. investigate the use of high-throughput sequencing (HTS) for direct identification of viral etiologies in individuals with or without central nervous system disease. This manuscript presents interesting findings which may add to the growing understanding of clinical utilization of NGS/HTS to aid diagnosis of infections, particularly in CNS infections with unknown causes. Of note, the inclusion of 30 specimens from patients with no known CNS disease provides the opportunity to explore the CSF virome. However, this manuscript may be benefit from revision. I would like to encourage the authors to address the comments and resubmit the manuscript.
Major comment1: Line 83, I would prefer the authors to clarify how positive control (monitor the nucleic acid recovery) and negative control (reagent control) were performed in this study and how these controls were used to help HTS analysis.
We thank the reviewer for this important comment. Indeed, the inclusion of positive and negative controls run in parallel to clinical samples is becoming highly recommended to evaluate the HTS analysis efficiency (e.g. sensitivity loss) as well as potential reagent/environmental contaminations (Miller S et al. Genome Res. 2019 May;29(5):831-842 ; Asplund M et al. Clin Microbiol Infect, https://doi.org/10.1016/j.cmi.2019.04.028). For the present study, the enrolment of patients with an initial suspicion of CNS inflammation started in May 2013 through June 2017. CSF samples were consecutively analysed by HTS, generating a large number of runs. Given costs constraints, positive (positive sample or spiked internal RNA/DNA controls) and negative controls run in parallel were not included. However, the HiSeq 2500 and 4000 platforms provided a huge sequencing depth (see Supplementary Figure 1), which should balance potential sensitivity losses. In contrast, HTS whole-process negative control was included for CSF samples collected from patients without CNS disease. This point represents a limitation of this study and is now clearly stated in the revised version.
Major comment 2: Line 99, the cell/chemistry analysis for the CSFs from the control group would add valuable information and help HTS data analysis from this group.
We agree with reviewer 2 that such data would have been of interest. However, it was challenging to obtain a CSF sample from patients not requiring lumbar puncture. Therefore, the limited material collected was dedicated to HTS analysis, and cell counts and chemistry analysis have not been performed.
Major comment 3: Line 134-135, please indicate if HTS identified these two infectious agents and how HTS results correlate with the diagnosis in these two cases.
Our pipeline is designed for viral sequences, and the Listeria monocytogenes case has thus not been confirmed by HTS in our study. Regarding the HIV-1 genome detected by real-time RT-PCR, it was not identified by HTS as mentioned in the manuscript. The copy number was low (100 copies/ml), and commercial HIV-1 rRT-PCR assays are known to be slightly more sensitive than HTS.
Major comment 4: Section 3.5,
The viral sequences assigned between two groups should be presented separately and included in the text so that readers would have a better overview. The data from each virus should be present in one paragraph/section to avoid confusion: For example, both line202 and 204 mentioned MCPyV, same as in line 204 and 206 for HpgV-1.
We are thankful for these comments, and we have rewritten this section according to the reviewer’s suggestions. We believe that these results are now shown with improved clarity.
Line 199 and 200, the numbers of TTV positive and negative sample should be included.
These numbers have been added, as requested.
Line 205, should be Figure S2, panel C.
We have checked: we indeed refer to Figure S2, panel A.
Major comment 5: The authors used specific r(RT)-PCR or RT-PCR assays to evaluate the trueness of the HTS findings (in line 113) and considered the identification of HHV7 from one control CSF as contamination (line 256). I would like to see the discussion of the rationale and compare the performance of HST vs. RT-PCR and cite relevant references.
We agree that we cannot rule out that in these cases HTS was more sensitive that the specific real-time PCR assays used for confirmation. This point is now mentioned in the revised version.
According to the reviewer’s comment, the performance of our unbiased HTS method compared to r(RT-)PCR assays but also to a positive selection HTS method (benchmarking with VirCapSeq-VERT from Lipkin WI group) is now discussed in the revised version. Indeed, our unbiased method has been challenged on different specimens (e.g. CSF, plasma, serum, bronchoalveolar lavage fluid, nasopharyngeal aspirate or swabs, stool, urine…) positive for a wide range of clinical relevant RNA and DNA viruses. The overall success rate suggested a sensitivity threshold close to that of real-time (RT-)PCR assays. Furthermore, HTS also made up for the lack of specificity of some diagnostic assays use for routine screening (Cordey S et al. Clin Microbiol Infect. 2015 Apr;21(4):387.e1-4).
Minor comment 1: Table 1, the bullets are not lined up.
This has been corrected.
Minor comment 2: Table 4, please include the normal ranges for the glucose and protein.
These normal ranges have been included.
Minor comment 3: Figure 2, please include the total number of samples for each group.
Figure 2 has been modified according to the reviewer’s comment.
Round 2
Reviewer 1 Report
The manuscript is now much improved and I recommend publication.
Reviewer 2 Report
The authors have made expanded effort in their revision. I recommend accepting the manuscript in the present form.
Thanks.